# Resistin Promotes Nasopharyngeal Carcinoma Metastasis through TLR4-Mediated Activation of p38 MAPK/NF-κB Signaling Pathway

**DOI:** 10.3390/cancers14236003

**Published:** 2022-12-05

**Authors:** Zongmeng Zhang, Jinlin Du, Qihua Xu, Yuyu Li, Sujin Zhou, Zhenggang Zhao, Yunping Mu, Allan Z. Zhao, Su-Mei Cao, Fanghong Li

**Affiliations:** 1The School of Biomedical and Pharmaceutical Sciences, Guangdong University of Technology, Guangzhou 510006, China; 2Department of Epidemiology and Health Statistics, School of Public Health, Guangdong Medical University, Dongguan 523808, China; 3Department of Cancer Prevention Research, Sun Yat-Sen University Cancer Center, Guangzhou 510060, China

**Keywords:** nasopharyngeal carcinoma, resistin, TLR4, metastasis, NF-κB

## Abstract

**Simple Summary:**

Chronic inflammation is associated with the development of nasopharyngeal carcinoma (NPC). Mounting evidence has indicated that resistin is an inflammatory cytokine that is associated with the risk of tumorigenesis. However, the correlation between serum resistin levels and the risk of NPC remains unclear. Here, we found that high serum resistin levels in NPC patients were positively correlated with lymph node metastasis and that resistin promoted the metastasis of NPC cells both in vitro and in vivo. Furthermore, we elucidated the underlying molecular mechanisms through which resistin promotes metastasis in NPC cells by inducing the epithelial-mesenchymal transition (EMT).

**Abstract:**

NPC is a type of malignant tumor with a high risk of local invasion and early distant metastasis. Resistin is an inflammatory cytokine that is predominantly produced from the immunocytes in humans. Accumulating evidence has suggested a clinical association of circulating resistin with the risk of tumorigenesis and a relationship between blood resistin levels and the risk of cancer metastasis. In this study, we explored the blood levels and the role of resistin in NPC. High resistin levels in NPC patients were positively associated with lymph node metastasis, and resistin promoted the migration and invasion of NPC cells in vitro. These findings were also replicated in a mouse model of NPC tumor metastasis. We identified TLR4 as a functional receptor in mediating the pro-migratory effects of resistin in NPC cells. Furthermore, p38 MAPK and NF-κB were intracellular effectors that mediated resistin-induced EMT. Taken together, our results suggest that resistin promotes NPC metastasis by activating the TLR4/p38 MAPK/NF-κB signaling pathways.

## 1. Introduction

Nasopharyngeal carcinoma (NPC) is a malignant tumor that originates from the nasopharyngeal epithelium [1]. The incidence of NPC is familial, with regional clustering existing in Southeast Asia and South China [2,3]. Established risk factors for NPC include Epstein–Barr virus (EBV) infection, a family history of NPC, and environmental factors [4,5,6]. In addition, the development of NPC is usually accompanied by chronic inflammation and metabolic dysregulation. Emerging evidence indicates that the immune system and cytokines may play an important role in the diagnosis and prognosis of NPC [7,8,9,10]. Indeed, our own previous study found that decreased levels of macrophage inflammatory protein (MIP)-1α and MIP-1β could increase the tumorigenic risk of NPC [11].

Inflammatory cytokines, such as adiponectin, resistin and leptin, have been shown to correlate with the development, progression and mortality of various types of cancer [12,13,14]. We recently showed that null mutations in adiponectin increase the occurrence of endometrial cancer in PTEN heterozygotic mutant mouse models [15]. Resistin is a peptide hormone that is predominantly synthesized and secreted from monocytes and macrophages in humans [16]. It has been implicated in promoting insulin resistance, obesity, chronic low-grade inflammation and tumor cell adhesion [17,18,19].

Recent studies have demonstrated that circulating resistin levels are significantly elevated in patients with breast, gastric, colorectal, lung and endometrial cancers [20,21,22,23,24], suggesting that serum resistin can be a potential diagnostic biomarker for cancers. Further studies have indicated that circulating levels of resistin are positively correlated with increased tumor stage, size and lymph node metastasis in various cancer subtypes [21,24,25] and that the resistin treatment could promote tumor cell proliferation, angiogenesis, migration and chemotherapy resistance in both animal models and cultured cells [26,27,28,29,30]. However, the correlation between serum resistin levels and the risk of NPC remains unclear.

In this study, we initially examined the clinical correlations of blood resistin levels with the risk of NPC in a case–control study and further explored the correlation between resistin levels and the clinical characteristics of NPC patients. Mechanistically, resistin promoted the migration and invasion of NPC cells in vivo and in vitro. Furthermore, we investigated the underlying mechanisms of a resistin-induced epithelial-mesenchymal transition (EMT) in NPC cells.

## 2. Materials and Methods

### 2.1. Patient Samples, Ethical Approval and Resistin Determination

The serum samples of 100 patients with NPC and 100 healthy controls undergoing routine health examinations were consecutively collected from the serum bank of Sun Yat-sen University Cancer Center (SYSUCC). The patients were selected based on previously described criteria [11]. This study was approved by the Institutional Review Board of Sun Yat-sen University Cancer Center (SYSUCC) (NO. YP2009051).

Serum resistin levels were measured using a Milliplex map kit and the Luminex 200TM instrument (Millipore, Billerica, MA, USA). The measurement procedure was strictly in accordance with the standard steps of the manufacturer’s kit, and the case and control serum samples were randomly distributed to each plate to ensure that the tester blindly detected the case–control state of the test sample.

### 2.2. Animal Study

All experimental procedures using animals were approved by the Experimental Animal Academic Ethics Committee of South China University of Technology (AEC2021059). For this study, 5-week-old male nude mice were purchased from GemPharmatech (Nanjing, Jiangsu, China) and were kept in a specific pathogen-free room in the Laboratory Animal Center of South China University of Technology (Guangzhou, Guangdong, China).

The model of lung metastasis due to the 5-8F-Luc cells stably expressing luciferase was established as previously described [31]. After the modeling, mice were injected intravenously with 20 μg/kg resistin for 2 weeks; first for three consecutive days, and then injections were administrated every other day. We detected 5-8F-Luc cell metastasis with the IVIS Lumina series Ⅲ imaging system (Xenogen, Alameda, CA, USA). After 6 weeks, the mice’s lungs were collected, weighed, and photographed. After fixing, paraffin embedding and sectioning, hematoxylin and eosin (HE), and immunohistochemistry staining were performed.

### 2.3. Cell Culture and Regents

The human NPC cell lines, CNE-2 and S18, were kindly gifted by Professor Chaonan Qian at SYSUCC. The HNE2 and 5-8F cell lines were obtained from the Central South University Advanced Research Center (Changsha, Hunan, China). The C666-1 cell line was obtained from the American Type Culture Collection (ATCC, Manassas, VA, USA). The CNE-2, C666-1 and S18 cells were cultured in Dulbecco’s modified eagle medium, and the HNE2 and 5-8F cells were cultured in the RPMI-1640 medium. All cells were supplemented with 10% fetal bovine serum (Gibco, Carlsbad, CA, USA) as well as 1% penicillin-streptomycin (Hyclone, Logan, UT, USA). The cells were maintained in a humidified atmosphere of 5% CO_2_ at 37 °C.

Recombination human resistin was dissolved in deionized water to prepare a working stock solution of approximately 0.01 mg/mL (PeproTech, Rocky Hill, NJ, USA). LPS-RS Ultrapure was purchased from InvivoGen (San Diego, CA, USA). SB203580, pyrrolidine dithiocarbamate ammonium (PDTC) and BAY 11-7082 were purchased from MedChem Express (Monmouth Junction, New Jersey, NJ, USA).

### 2.4. Cell Viability and Proliferation Assays

The NPC cells were cultured in 96-well plates and were treated with different concentrations of resistin for 48 h. After the incubation of the CCK-8 solution was carried out according to the provided instructions (Sangon Biotech, Shanghai, China), the absorbance was measured at 450 nm with a microplate reader (Infinite F50, Tecan Group Ltd., Mannedorf, Switzerland). The relative cell viability was calculated as the percentage of untreated cells. Cell proliferation was measured using plate clone formation and was carried out as previously described [32].

### 2.5. Wound-Healing Assay

The wound-healing assay was performed as previously described [31]. The NPC cells were plated in 12-well culture plates, and cell confluence was 100% after adherence. After the monolayer cells were scraped with a plastic 200 µL pipette tip, the cells were washed with phosphate-buffered saline (PBS) and incubated with fresh medium. The cell-scratch images were photographed using a microscope at 0 h and 24 h. The relative migration rates were calculated using the following calculation: cell-covered area (0 h)/cell-covered area (24 h).

### 2.6. Migration and Invasion Assays

The migration and invasion assays were performed using 24-well Transwell inserts (BD Biosciences, San Jose, CA, USA) coated with or without growth factor-reduced Matrigel (Corning Incorporated, Corning, NY, USA), as previously described [31]. The cells were resuspended in a 200 ul serum-free medium that had been treated with or without resistin, which was added to the Transwell’s upper chamber, and a medium with 20% FBS was added to the Transwell’s lower chamber. For the signaling blockade, cells were pre-incubated with an inhibitor for 2 h. After 24 h of incubation, membrane-trapped cells were fixed with 4% paraformaldehyde, stained with 10% crystal violet solution and counted using a light microscope.

### 2.7. Transient Transfection with Small Interfering RNA (siRNA)

TLR4, p38 MAPK and scrambled control siRNAs were synthesized by RiboBio (Guangzhou, Guangdong, China) and transfected using a transfection reagent kit (RiboBio) according to the manufacturer’s protocols. The siRNA sequences used for this study are listed in Appendix A.

### 2.8. RNA Extraction and qRT-PCR

RNA samples were extracted with Trizol reagent (Sigma-Aldrich, St. Louis, MO, USA). A cDNA synthesis was performed with the HiScript II Q RT kit (Vazyme Biotech, Nanjing, China). The quantitative real-time PCR (qRT-PCR) analysis was performed as previously described [32]. Primers were synthesized by Sangon Biotech (Shanghai, China). The primer sequences are listed in Appendix A.

### 2.9. Western Blot Analysis

A Western blot analysis was performed as previously described [31]. Briefly, protein samples were extracted with radio-immunoprecipitation (RIPA) lysis buffer containing a protease inhibitor (Beyotime Biotechnology, Shanghai, China) and quantified with the bicinchoninic acid (BCA) protein assay kit (Beyotime Biotechnology). Cytosolic and nuclear proteins were extracted using the Nuclear and Cytoplasmic Protein Extraction Kit according to the manufacturer’s instructions (Beyotime Biotechnology). Protein samples were separated using SDS-PAGE and were transferred to a polyvinylidene fluoride (PVDF) membrane (Millipore, Billerica, MA, USA). The membranes were blocked with 5% nonfat dried milk and incubated overnight at 4 °C with the primary antibody and then incubated with the secondary antibody for 1 h at room temperature. The blots were tested with the ECL detection system (Thermo, Waltham, MA, USA) using the ChemiDoc XRS+ system (Bio-Rad, Hercules, CA, USA). The antibodies used are listed in Appendix A.

### 2.10. Immunofluorescence Staining

Immunofluorescence staining was carried out as previously reported [31]. The NPC cells were cultured onto glass-bottom cell culture dishes (Wuxi NEST Biotechnology Co., Ltd., Jiangsu, China). After incubation with resistin, the cells were fixed with 4% paraformaldehyde, permeabilized in 0.2% Triton X-100 PBS buffer, and then blocked with Immunol Staining Blocking Buffer (Beyotime), incubated with primary antibody rabbit anti-p65 (CST; 1:400) overnight at 4 °C, and then incubated with goat anti-rabbit Alexa 555 fluorescent secondary antibody (CST; 1:1000) for 2 h at room temperature. After washing with PBS, the cells were mounted with an anti-fade solution with DAPI (Beyotime). The images were obtained using a fluorescent microscope.

### 2.11. Dual-Luciferase Reporter Assay

The pNFκB-luc, pRL-TK plasmids, and dual-luciferase reporter assay kit were purchased from Beyotime. The dual-luciferase reporter assay was determined as previously reported [31] using a Varioskan LUX multimode microplate reader (Thermo). Relative luminescence units = Firefly luciferase activity/Renilla luciferase activity.

### 2.12. Immunohistochemistry Staining

Immunohistochemistry staining was carried out as described elsewhere [31]. After dewaxing and rehydrating, microwave heating for antigen retrieval was performed on the sections in a citrate antigen retrieval solution. After blocking with 3% H_2_O_2_ for 15 min, the sections were incubated with 5% goat serum buffer for 1 h at room temperature, followed by overnight incubation at 4 °C with the primary antibodies mouse anti-p-p65 (CST; 1:50), rabbit anti-Vimentin (CST; 1:50) and rabbit anti-E-cadherin (CST; 1:100). Then, the sections were incubated with the secondary antibody for 1 h at room temperature. The sections were incubated with a developing solution (diaminobenzidine, DAB) and counterstained with hematoxylin (Wuhan Servicebio Technology, Hubei, China).

### 2.13. Statistical Analysis

In order to describe the cohort characteristics, the Chi-square (χ2) test and Wilcoxon rank-sum test were used to compare the differences between the case and control groups. The median levels of resistin among the cases and controls were compared using the Wilcoxon rank-sum test to compare the differences between groups. In the multivariable models, we adjusted for the established or suspected risk factors of NPC, including age, gender, EBV VCA-IgA and EBNA1 IgA. The odds ratios (ORs) and 95% confidence intervals (95% CIs) for the risk of NPC were computed using an unconditional logistic regression model.

The differences in the serum resistin levels among NPC patients with different clinical characteristics were analyzed using the Wilcoxon rank-sum test. The correlation of the serum resistin levels with LN metastasis was analyzed by unconditional logistic regression. Statistical analyses were performed using the SAS statistical software, version 9.4 (SAS Institute, Cary, NC, USA). All hypothesis tests underwent two-sided testing, and a *p* < 0.05 was considered statistically significant.

Data are presented as mean ± SD. All the data (except clinical) were analyzed using a Student’s *t*-test or one-way ANOVA with Sidak’s multiple comparisons test using GraphPad Prism 8.0 (GraphPad Software, La Jolla, CA, USA). A value of *p* < 0.05 was considered statistically significant.

## 3. Results

### 3.1. Clinical Correlation of Serum Resistin Levels with the Risk of NPC

The descriptive characteristics of the study subjects with NPC and the controls who provided serum are presented in Table 1. The serum resistin levels were significantly higher in the NPC patients compared with the controls (4.12 vs. 3.59 ng/mL; *p* < 0.001) (Table 1). The sex-corrected resistin levels were higher among the cases compared to the controls; these differences were statistically significant among men (4.18 vs. 3.58 ng/mL; *p* < 0.001) and women (4.38 vs. 3.46 ng/mL; *p* = 0.003) (Table 1).

In multivariable logistic regression models, we observed that high serum resistin levels were associated with increased NPC risk after adjusting for established or suspected risk factors of NPC, including age, gender, EBV VCA-IgA and EBNA1 IgA (Table 2). Through analyzing the differences in serum resistin levels with clinical characteristics, we observed different levels of serum resistin in NPC patients with different types of lymph node metastasis (Table 3). Moreover, the serum resistin level was a significant independent predictor for lymph node metastasis in NPC patients, according to multivariate logistic regression analysis after adjusting for established or suspected risk factors of NPC, including age, gender, EBV VCA-IgA and EBNA1 IgA (Table 4).

### 3.2. Resistin Does Not Affect the Proliferation but Promotes the Migration and Invasion in NPC Cells

To determine the manifestations of the clinical correlation of resistin, we further explored whether resistin affected the activity of NPC cells. A co-culture of different concentrations of resistin for 48 h with NPC cells did not affect cell viability (Figure 1A). Furthermore, resistin treatment did not affect the proliferation of the NPC cells in a colony formation assay after long-term incubation (Figure 1B).

Interestingly, resistin treatment of the NPC cells enhanced wound healing and the migration and invasion activities in a dose-dependent manner (Figure 2A–C). The epithelium-mesenchymal transition (EMT) plays an important role in tumor cell invasion and cancer metastasis [33]. We found that resistin also induced the expression of EMT-promoting transcription factors, such as ZEB1 and Snail and Slug, via the Western blot assays; however, the level of β-catenin was not altered by the resistin treatment (Figure 2D). The loss of E-cadherin expression is a hallmark of an EMT [34]. The resistin treatment significantly suppressed the expression of E-cadherin in NPC cells as well as other epithelial markers, such as claudin-1 and ZO-1 (Figure 2D). Conversely, the levels of N-cadherin and vimentin, which are the hallmarks of mesenchymal cells, were significantly increased after the resistin treatment (Figure 2D). Importantly, resistin also elevated the expression of matrix metalloproteinase 2 (MMP-2) and matrix metalloproteinase 9 (MMP-9) (Figure 2D), both of which are essential for cell motility and invasion. Thus, resistin can promote migration and invasion by inducing an EMT in NPC cells.

### 3.3. TLR4 Is Necessary for Resistin-Induced NPC Cell Migration

Resistin is a type of cysteine-rich polypeptide hormone that functions through its purported receptor, Toll-like receptor 4 (TLR4), which plays a critical role in regulating inflammation and is also involved in tumor cell proliferation, invasion and metastasis [25,28,29]. The expression of TLR4 is widely observed in head and neck squamous cell carcinoma (HNSC) and NPC tissues (Appendix A), as well as in several NPC cell lines (Appendix A). Although there is no discernable difference in TLR4 expression in NPC, as compared with the normal tissues (Appendix A), high levels of TLR4 expression were correlated with increasing tumor grade and nodal metastasis (Appendix A). The blockade of TLR4 with the pharmaceutical inhibitor LPS-RS Ultrapure, a specific TLR4 antagonist, suppressed cell migration and invasion induced by the resistin treatment (Figure 3A–D). Similarly, the knockdown of TLR4 expression significantly nullified the resistin-induced elevation of N-cadherin, MMP-2 and MMP-9 expression, as well as a reduction of E-cadherin in the NPC cells (Figure 3E and Appendix A). These results unequivocally demonstrate that resistin induces the migration and invasion of NPC cells through TLR4.

### 3.4. The p38 MAPK Signaling Pathway Is Involved in Resistin-Induced Migration in NPC Cells

We further examined the downstream signaling events of TLR4. While resistin did not affect the phosphorylation of AKT, it stimulated the phosphorylation of p38 mitogen-activated protein kinase (MAPK) (Figure 4A) and suppressed the level of ERK1/2 phosphorylation (Figure 4A). Incubation of the NPC cells with a specific p38 inhibitor, SB203580, largely reversed resistin-induced migration (Figure 4B), whereas the inhibitors of ERK1/2, JNK and AKT showed no effect on the migration (Figure 4B). Consistent with this observation, siRNA’s knockdown of p38 MAPK expression prevented cell migration and invasion in the resistin-treated NPC cells (Figure 4C,D and Appendix A). Due to the reduction in p38 MAPK expression, the resistin-induced changes in the EMT-related proteins were inhibited by transfection with p38 MAPK siRNA (Figure 4E). In analyzing the whole cell lysates from resistin-treated cells, we also found that blocking TLR4 activity via LPS-RS Ultrapure or siRNA transfection abolished the resistin-induced activation of p38 MAPK (Figure 4F,G), further proving that TLR4 was required for the resistin-induced activation of p38 MAPK in NPC cells.

### 3.5. Resistin Regulates Expression of EMT-Related Protein via NF-κB

The involvement of Nuclear factor-κB (NF-κB) in regulating the expression of EMT-related protein has been well documented [35,36]. Co-cultures with resistin increased the phosphorylation of the p65 protein in a dose-dependent manner in the NPC cells (Figure 5A). The resistin treatment promoted the phosphorylation of IκBα (Figure 5B), which, in turn, led to the degradation of IκBα and to the activation of NF-κB. Consistent with these results, the proportion of nuclear translocation of the p65 and p50 proteins markedly increased following the resistin treatment (Figure 5C). Moreover, pretreatment with the NF-κB inhibitors, BAY-117083 and PDTC, completely suppressed a resistin-induced EMT as well as the migration of NPC cells (Figure 5D,E). These results demonstrate that resistin promotes the EMT of NPC cells, largely through the activation of the NF-κB pathway.

We further delineated the molecular mechanisms underlying the resistin-induced EMT alterations in the NPC cells, particularly with respect to NF-κB signaling. The transcriptional activation of NF-κB, induced by resistin, was suppressed by co-culturing with LPS-RS Ultrapure, a specific inhibitor of TLR4 (Figure 6A). The immunofluorescence staining revealed that the resistin-induced nuclear translocation of p65 was abolished by the LPS-RS Ultrapure treatment in the NPC cells (Figure 6B). Importantly, by impeding the activation of p38 MAPK via its pharmacological inhibitor, it also suppressed the resistin-mediated activation of NF-κB in the NPC cells (Figure 6A,C) and reversed the resistin-induced phosphorylation of IκBα (Figure 6D,E). These results, taken together, demonstrate that the induction of cellular migration by resistin depends on the TLR4/p38 MAPK/NF-κB signaling pathways.

### 3.6. Resistin Promotes NPC Tumor Metastasis in Animal Models

To clarify whether intravenously administered resistin would exhibit a pharmacokinetic profile suitable for an in vivo evaluation, we measured the serum concentrations of resistin in nude mice after the intravenous administration of 20 μg/kg of resistin. We found that the concentration of resistin was 20.79 ng/mL at 15 min (Figure 7A), which was consistent with the concentration of resistin that promoted migration in vitro.

To understand the effects of resistin in metastasis in vivo, we established luciferase-expressing 5-8F-Luc cells. A lung metastasis model was established by intravenously injecting 5-8F-Luc cells into nude mice, and the tumor metastasis was monitored by bioluminescence imaging (Figure 7B). The intravenous delivery of the exogenous recombinant resistin proteins significantly increased lung metastasis at 6 weeks post-injection (Figure 7C–E), with the lungs showing more and larger metastatic nodules in the resistin-treated group than in the control group (Figure 7F,H), accompanied by the lungs exhibiting an increased wet weight (Figure 7G). The immunohistochemical staining showed that treatment with exogenous resistin markedly elevated the levels of phospho-p65 and vimentin (Figure 7I,J). These results, taken together, unequivocally validated the concept that elevation of blood resistin could enhance the metastasis ability of NPC cells in the metastatic animal model.

## 4. Discussion

Using a case–control cohort of 100 patients and 100 controls, we revealed, for the first time, that high serum resistin levels were associated with an increased risk of NPC. Importantly, we showed that the serum resistin levels were positively correlated with lymph node metastases in NPC patients. Consistent with these clinical findings, the resistin treatment promoted the invasion and migration of NPC cells in cultured cells as well as metastasis in a human NPC cell-derived animal model. Resistin promoted the invasion and migration of NPC cells by inducing an EMT, a molecular event that was initiated by the interaction of resistin with its purported receptor, TLR4, and further mediated by the activation of the p38 MAPK and NF-κB pathways.

Nasopharyngeal carcinoma is typically characterized by heavy lymphocytic infiltration, suggesting that inflammation might be a potential risk factor for the progression of this cancer [11]. Indeed, a series of cytokines, such as leptin, adiponectin and visfatin, have been found in tumor microenvironments and have been implicated in cancer cell growth, apoptosis, invasion, angiogenesis and metastasis [13,19]. Resistin is a cytokine that is predominantly produced and secreted by macrophages, dendritic cells and monocytes in humans [17,37]. The purported ortholog receptor of resistin, TLR4, is usually expressed and has recently been identified on multiple tumor cells, including gastric cancer, breast cancer and lung adenocarcinoma [25,28,29]. Recent studies have already suggested that the polymorphisms and high expression of TLR4 are linked to an increased risk of NPC [38,39,40]. Our results indicate that TLR4 is widely expressed in NPC as well as in head and neck tumors; that its expression level is positively correlated with high grades of tumor and lymph node metastasis in HNSC; that the inhibition of TLR4 signaling prevents resistin-induced migration and invasion; and that TLR4 knockdown prevents the resistin-induced expression of multiple critical EMT proteins. These findings are consistent with published reports, indicating that the activation of TLR4 could promote cancer cell proliferation, adhesion, EMT, invasion and migration [25,28,29]. Thus, TLR4 is the functional receptor of resistin signaling and is responsible for mediating the pro-metastatic effect of resistin in NPC cells.

Intracellular signaling pathways, such as MAPK and PI3K/AKT, are involved in mediating TLR4 functions [41,42]. In NPC cells, we only found the activation of p38 MAPK signaling after resistin treatment and that pretreatment with specific inhibitors of p38 MAPK largely reversed resistin-induced migration or invasion. Importantly, the blockade of TLR4 signaling reduced the resistin-induced activation of p38 MAPK signaling, proving that the TLR4/p38 MAPK signaling pathways are critical for resistin’s induction of migration and invasion. The NF-κB proteins belong to a family of transcription factors that are involved in cellular functions, such as inflammation, immune responses, cell proliferation and apoptosis [41]. Moreover, NF-κB is an important regulator of the EMT process of tumor cells [35,36]. The activation of NF-κB by the cytokines from the tumor microenvironment plays an important role in the invasion and migration of NPC cells [43]. Indeed, pharmacological inhibition of the NF-kB signaling pathways attenuates resistin-induced EMT-related protein expression, a process that depends on the activation of the TLR4/p38/NF-κB pathway (Figure 8). These data provide an underlying mechanism describing how high blood levels of resistin promote the metastasis of NPC.

## 5. Conclusions

In conclusion, the findings of this study demonstrate that serum resistin levels are positively correlated with the risk of NPC development and can potentially serve as an independent predictor of lymph node metastasis in NPC cases. We propose that resistin promotes NPC metastasis through the induction of an EMT by activating the TLR4/p38 MAPK /NF-κB signaling pathways. The circulating levels of resistin may be considered for predicting the prognosis of NPC patients.

## Figures and Tables

**Figure 1 cancers-14-06003-f001:**
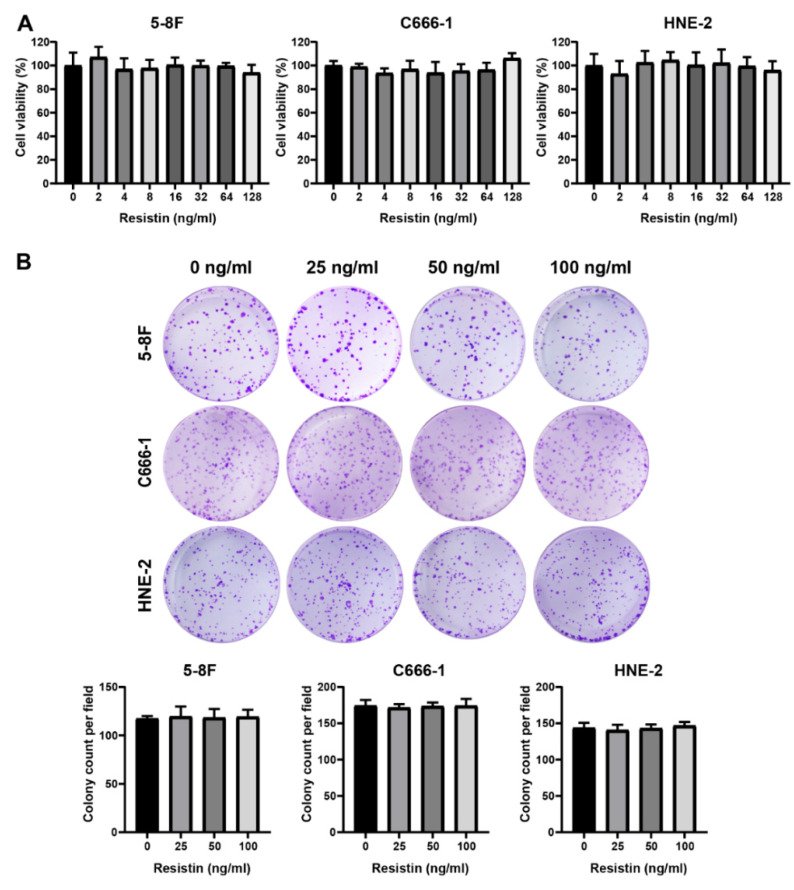
Resistin does not affect proliferation in NPC cells. (**A**) The viability of HNE-2, 5-8F and C666-1 cells was determined after treatment with different doses of resistin for 48 h. (**B**) The clonogenic ability of HNE-2, 5-8F and C666-1 cells was evaluated after treatment with different concentrations of resistin (0, 25 or 50 ng/mL) for 7 days. Graphs show relative colony numbers. Results are presented as mean ± SD of three independent experiments performed in triplicate.

**Figure 2 cancers-14-06003-f002:**
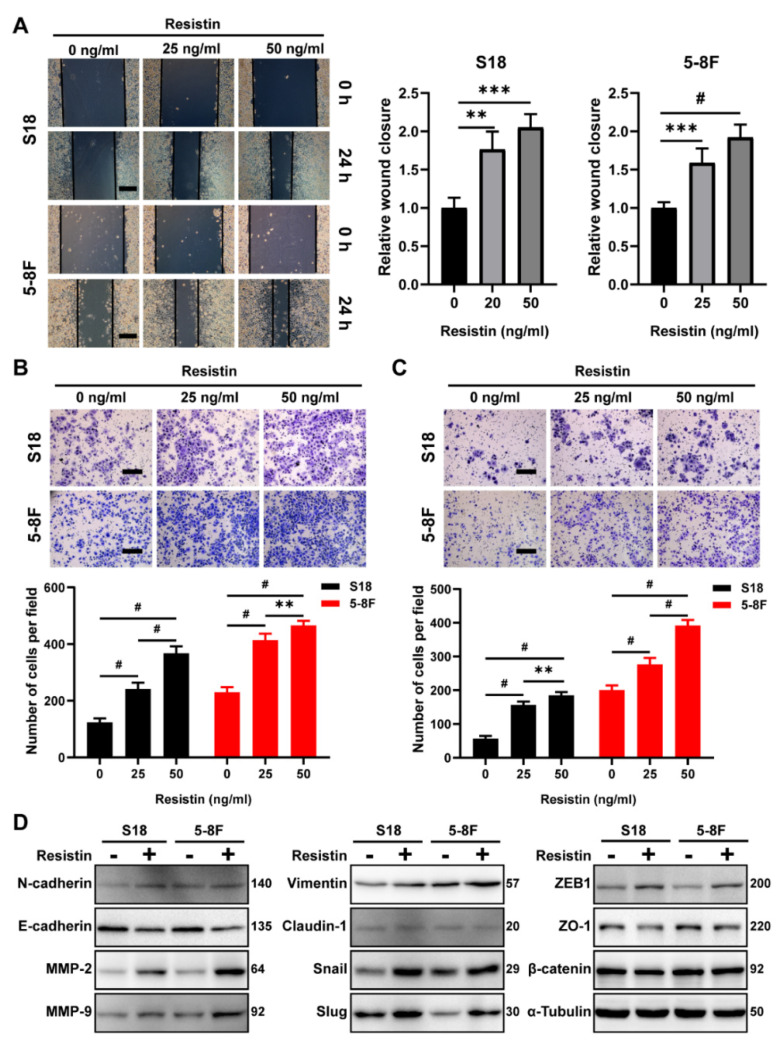
Resistin promotes invasion and migration in NPC cells. (**A**) Wound−healing assay of S18 and 5-8F cells pretreated with different concentrations of resistin (0, 25 or 50 ng/mL) for 24 h. Scale bar: 200 μm. Graphs show the relative wound closure. (**B**) Migration and (**C**) invasion of S18, 5-8F cells were evaluated by Transwell assay after treatment with different concentrations of resistin (0, 25 or 50 ng/mL) for 24 h. Scale bar: 200 μm. Graphs show the relative number of migratory and invasive cells. (**D**) Western blot analysis of E-cadherin, N-cadherin, MMP-2, MMP-9, Slug, Snail, Vimentin, Claudin-1, ZEB-1, ZO-1 and β-catenin in cultured S18 and 5-8F cells after treatment with or without 25 ng/mL resistin for 24 h. Results are presented as mean ± SD of three independent experiments performed in triplicate. ** *p* < 0.01, *** *p* < 0.001, ^#^ *p* < 0.0001.

**Figure 3 cancers-14-06003-f003:**
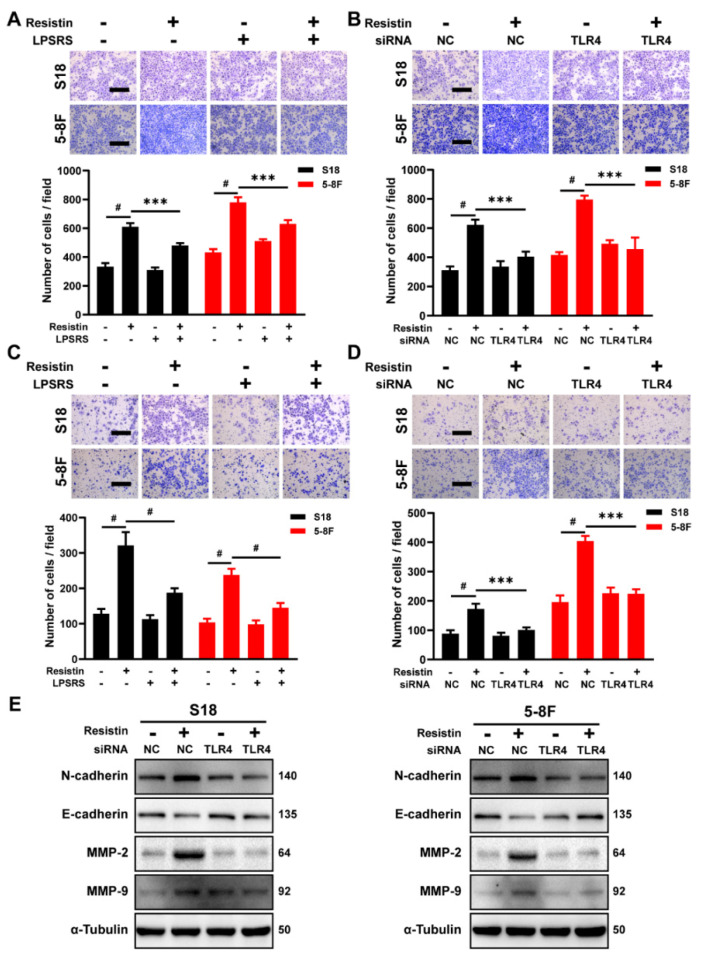
TLR4 is the functional receptor of resistin in NPC cells. (**A**) Transwell assay of migration activities in NPC cells incubated with 25 ng/mL resistin after pretreatment with or without 5 μg/mL LPS-RS Ultrapure (TLR4 inhibitor). Scale bar: 200 μm. Graphs show the relative number of migration cells. (**B**) NPC cells were transfected with NC or TLR4 siRNA. Transwell assay of migration activities in NPC cells incubated with 25 ng/mL resistin after 24 h of transfection. Scale bar: 200 μm. Graphs show the relative number of migration cells. (**C**) Transwell assay of invasion activities in NPC cells incubated with 25 ng/mL resistin after pretreatment with or without 5 μg/mL LPS-RS Ultrapure (TLR4 inhibitor). Scale bar: 200 μm. Graphs show the relative number of invasion cells. (**D**) NPC cells were transfected with NC or TLR4 siRNA. Transwell assay of invasion activities in NPC cells incubated with 25 ng/mL resistin after 24 h of transfection. Scale bar: 200 μm. Graphs show the relative number of invasion cells. (**E**) S18 and 5-8F cells were transfected with NC or TLR4 siRNA, and E-cadherin, N-cadherin, MMP-9 and MMP-2 levels were then determined by Western blot analysis after treatment with resistin (25 ng/mL) for 24 h. Results are presented as mean ± SD of three independent experiments performed in triplicate. *** *p* < 0.001, ^#^
*p* < 0.0001.

**Figure 4 cancers-14-06003-f004:**
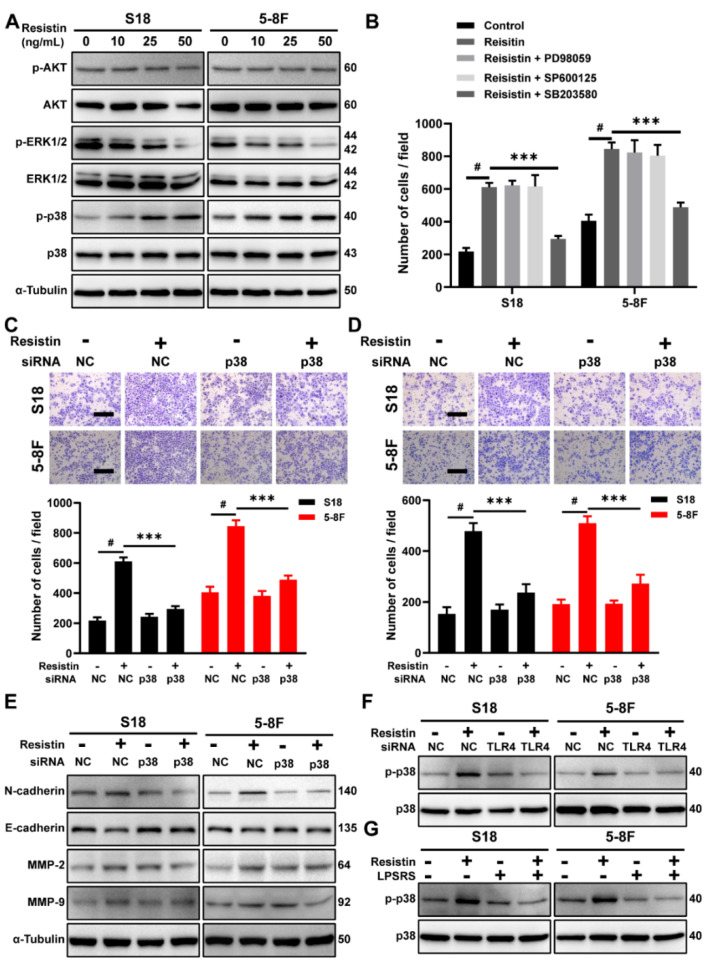
The p38 MAPK pathway is required for resistin−induced migration in NPC cells. (**A**) Western blot analysis of AKT, p--AKT, p38 MAPK, p-p38 MAPK, ERK1/2 and p-ERK1/2 in cultured S18 and 5-8F cells after treatment with different concentrations of resistin (0, 10, 25 or 50 ng/mL) for 1 h. (**B**) Transwell assay of migration activities in NPC cells incubated with 25 ng/mL resistin after pretreatment with vehicle or various inhibitors (10 μM SB203580, 10 μM SP600125 or 10 μM U0126). NPC cells were transfected with NC or p38 MAPK siRNA. After 24 h of transfection, Transwell assays of (**C**) migration and (**D**) invasion activities in NPC cells incubated with 25 ng/mL resistin. Scale bar: 200 μm. Graphs show the relative number of migratory and invasive cells. (**E**) S18 and 5-8F cells were transfected with NC or p38 MAPK siRNA, E-cadherin, N-cadherin, MMP-9, and MMP-2 protein levels were then determined by Western blot analysis after treatment with resistin (25 ng/mL) for 24 h. (**F**) S18 and 5-8F cells were transfected with NC or TLR4 siRNA. After 24 h of transfection, p38 MAPK and p-p38 MAPK protein levels were then determined by Western blot analysis after treatment with resistin (25 ng/mL) for 1 h. (**G**) NPC cells were pretreated with LPS-RS Ultrapure; p38 MAPK and p-p38 MAPK protein levels were then determined by Western blot analysis after treatment with resistin (25 ng/mL) for 1 h. Results are presented as mean ± SD of three independent experiments performed in triplicate. *** *p* < 0.001, ^#^
*p* < 0.0001.

**Figure 5 cancers-14-06003-f005:**
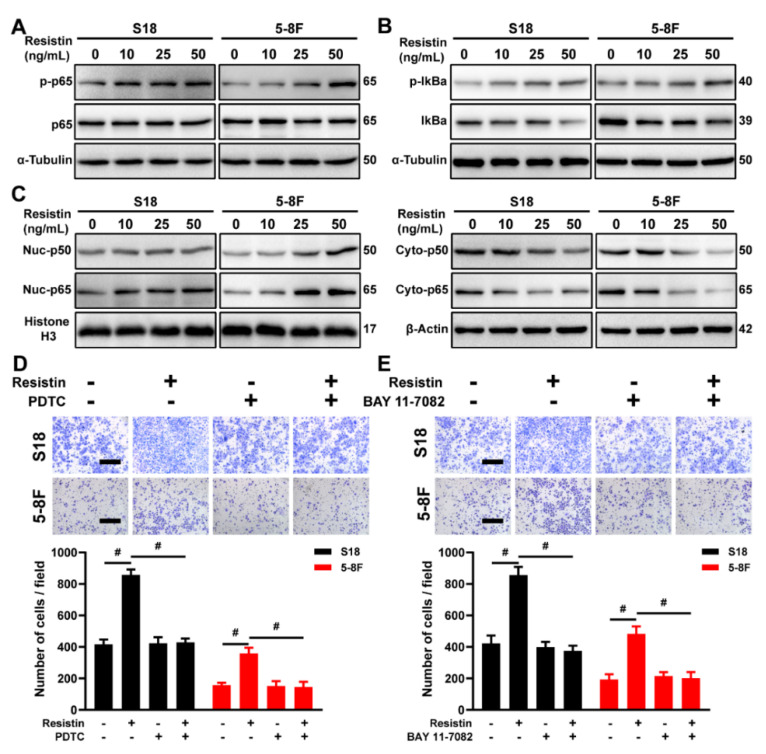
NF-κB is the downstream signal mediating resistin−induced migration and invasion of NPC cells. (**A**) Western blot analyses of p65 and p-p65 in cultured S18 and 5-8F cells after treatment with different concentrations of resistin (0, 10, 25 or 50 ng/mL) for 1 h. (**B**) Western blot analyses of IκBα and p-IκBα in cultured S18 and 5-8F cells after treatment with different concentrations of resistin (0, 10, 25 or 50 ng/mL) for 1 h. (**C**) Western blot analyses of cyto-p50, cyto-p65, nuc-p50 and nuc-p65 in cultured S18 and 5-8F cells after treatment with different concentrations of resistin (0, 10, 25 or 50 ng/mL) for 12 h. Transwell assay of migration activities in NPC cells incubated with 50 ng/mL resistin after pretreatment with or without 50 μM PDTC (**D**) or 10 μM BAY 11-7082 (**E**). Scale bar: 200 μm. Graphs show the relative number of migratory cells. Results are presented as mean ± SD of three independent experiments performed in triplicate. ^#^
*p* < 0.0001.

**Figure 6 cancers-14-06003-f006:**
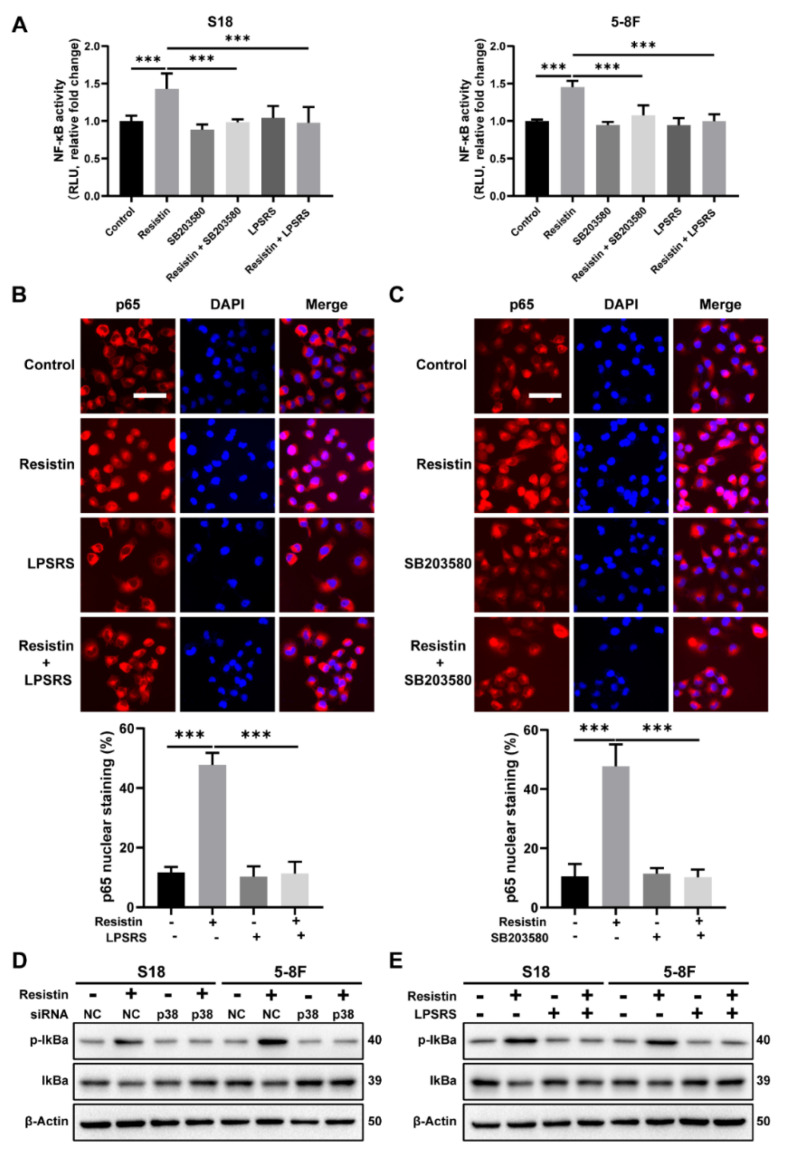
TLR4 and p38 MAPK signaling pathways are involved in resistin−induced NF-κB transcriptional activity. (**A**) The NF-κB reporter assay was performed to monitor the effects of resistin on NF-κB−regulated signal transduction pathways. NPC cells were co-transfected with the firefly and Renilla luciferase reporter. After transfection, cells were pretreated with LPS-RS Ultrapure (5 μg/mL) or SB203580 (10 μM) for 2 h and followed by resistin treatment for 6 h (25 ng/mL). (**B**) The distribution of NF-κB p65 and p50 subunits was visualized by immunofluorescent staining in NPC cells incubated with 25 ng/mL resistin after pretreatment with or without LPS-RS Ultrapure. Scale bar: 20 μm. (**C**) The distribution of NF-κB p65 and p50 subunits was visualized by immunofluorescent staining in NPC cells incubated with 25 ng/mL resistin after pretreatment with or without SB203580. Scale bar: 20 μm. (**D**) S18 and 5-8F cells were transfected with NC or p38 MAPK siRNA. After 24 h of transfection, IκBα and p-IKBα protein levels were then determined by Western blot analysis after treatment with resistin (25 ng/mL) for 1 h. (**E**) NPC cells were pretreated with LPS-RS Ultrapure, and then the IκBα and p-IκBα protein levels were determined by Western blot analysis after treatment with resistin (25 ng/mL) for 1 h. Results are presented as mean ± SD of three independent experiments performed in triplicate. *** *p* < 0.001.

**Figure 7 cancers-14-06003-f007:**
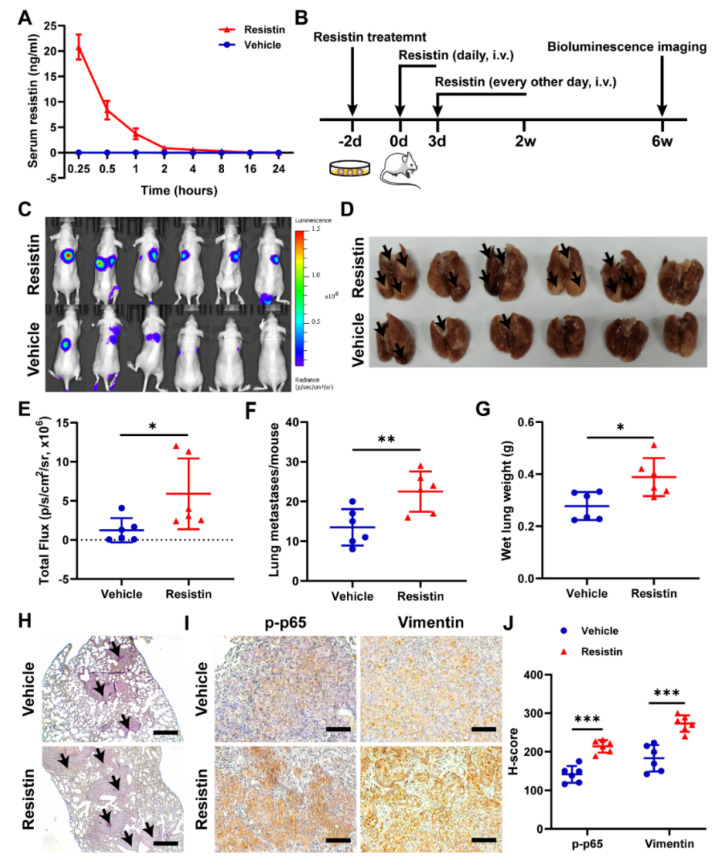
Resistin promotes NPC cell metastasis in vivo. (**A**) After resistin (20 μg/kg) was administered intravenously, the concentration of recombinant human resistin in mouse (*n* = 3) serum was detected by ELISA. (**B**) 5-8F-Luc cells (1 × 10^6^ cells per mouse) were treated with or without resistin (25 ng/mL) for 2 d and then injected into the tail veins of male nude mice (*n* = 6). Mice were injected intravenously with 20 μg/kg resistin for two weeks for the first three consecutive days and then received injections every other day. (**C**) In vivo photon emission of 5-8F-Fluc cells in lungs was detected and photographed by the IVIS50 image system at week 6. (**E**) Quantification of in vivo bioluminescence imaging images (photons/s of lung region). (**D**,**F**,**G**) After 6 weeks, the mice were sacrificed, and the lung tissues were excised and photographed. Wet lungs were counted and summarized. (**H**) According to the HE staining of the lungs, the number of lung metastases was counted and summarized. Scale bar: 200 μm. (**I**,**J**) Immunohistochemistry of p-p65 and vimentin protein expressions in metastatic NPC cells from lungs. Scale bar: 100 μm. Results are presented as mean ± SD of three independent experiments performed in triplicate. * *p* < 0.05, ** *p* < 0.01, *** *p* < 0.001.

**Figure 8 cancers-14-06003-f008:**
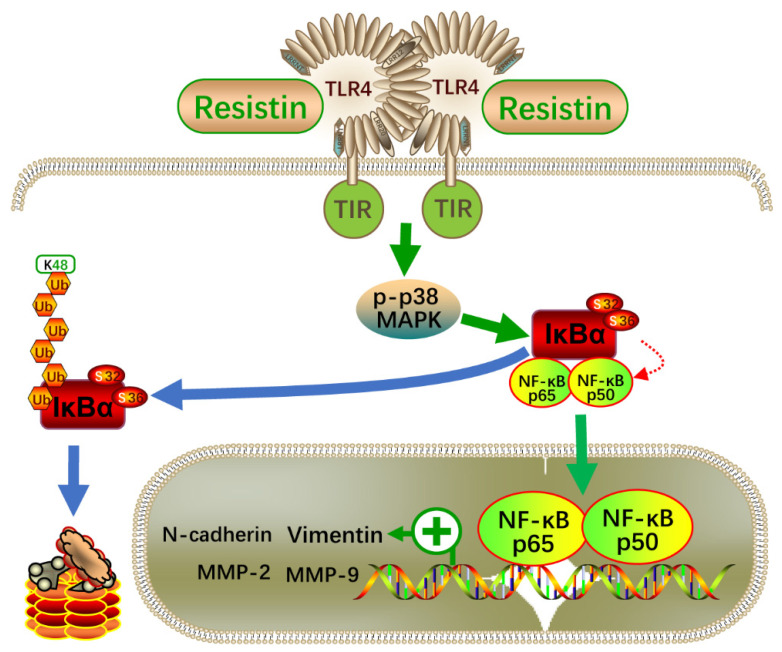
Schematic illustration of the resistin-induced migration and invasion in NPC cells. NF-κB—nuclear factor-κB; TLR4—Toll-like receptor 4; p38—p38 mitogen-activated protein kinase (MAPK); IκB—an inhibitor of κB.

**Table 1 cancers-14-06003-t001:** Characteristics of nasopharyngeal carcinoma cases and control subjects.

Characteristic	Case	Control	*Z/χ* ^2^	*p* ^a^
Total	100	100		
Gender				
Male	87	84	0.363	0.547
Female	13	16		
Age (M, IQR)	44.5 (38.0–52.0)	45.0 (38.0–54.0)	−0.444	0.657
Age group				
≤40	34	34	0.000	1.000
>40	66	66		
VCA-IgA				
Negative	23	46	11.705	0.001
Positive	77	54		
EBNA1				
Negative	32	78	42.747	<0.001
Positive	68	22		
Resistin ^b^ (M, IQR)				
Total	4.12 (3.67–4.50)	3.59 (3.11–3.98)	−6.998	<0.001
Female	4.38 (3.65–4.89)	3.46 (3.67–4.00)	−2.954	0.003
Male	4.18 (3.67–4.48)	3.58 (3.67–3.98)	−6.347	<0.001

Abbreviations: M—median; IQR—interquartile range; VCA-IgA—viral-capsid antigen–IgA; EBV—Epstein–Barr virus. ^a^
*p* values were calculated by Wilcoxon rank-sum and Chi-square tests. ^b^ Resistin concentration was described by median (ng/mL) and interquartile range.

**Table 2 cancers-14-06003-t002:** Analysis of multivariate logistic regression with risk for NPC ^a^.

Variable	b	SE	Wald	*p*	OR	OR 95%CI
Resistin	2.605	0.435	35.871	<0.001	13.531	5.769–31.735
EBNA1	1.950	0.393	24.582	<0.001	7.031	3.252–15.199
VCA-IgA	0.982	0.401	6.013	0.014	2.670	1.218–5.854
Constant	−11.578	1.839	39.630	<0.001	-	-

Abbreviations: OR—odd ratio; 95% CI—95% confidence interval; VCA-IgA—viral-capsid antigen–IgA; EBV—Epstein–Barr virus. ^a^ Multivariate analysis was performed by unconditional logistic regression models with a constant as the model intercept.

**Table 3 cancers-14-06003-t003:** Differences in serum resistin among NPC patients with different clinicopathological features.

Characteristic	*n*	Resistin ^a^ (M, IQR)	*Z*	*p* ^b^
LN metastasis				
No	68	4.06 (3.62–4.41)	−2.949	0.003
Yes	32	4.35 (4.04–4.59)		
Recurrence				
No	94	4.20 (3.66–4.48)	−0.501	0.616
Yes	6	4.24 (3.53–4.98)		
Stage ^c^				
Early (Ⅰ/Ⅱ)	36	4.12 (3.54–4.53)	−1.009	0.313
Advance (Ⅲ/Ⅳ)	64	4.24 (3.73–4.49)		
Gender				
Male	87	4.18 (3.67–4.48)	−1.164	0.245
Female	13	4.38 (3.65–4.89)		
Age group	100			
≤40	34	4.29 (3.74–4.49)	−0.786	0.432
>40	66	4.16 (3.64–4.50)		

Abbreviations: M—median; IQR—interquartile range; LN—lymph node. ^a^ Resistin concentrations were described by the median (ng/mL) and interquartile range. ^b^
*p* values were calculated by Wilcoxon rank-sum test. ^c^ Stage was based on the AJCC TNM staging system.

**Table 4 cancers-14-06003-t004:** Analysis of multivariate logistic regression with risk for NPC LN metastasis as a dependent variable ^a.^

Variable	b	SE	Wald	*p*	OR	OR 95%CI
Resistin	1.697	0.560	9.176	0.002	5.460	1.821–16.374
EBNA1	−0.006	0.517	0.000	0.990	0.994	0.361–2.736
VCA-IgA	0.876	0.637	1.889	0.169	2.401	0.689–8.374
Constant	−8.534	2.593	10.829	0.001	-	-

Abbreviations: OR—odd ratio; 95% CI—95% confidence interval; VCA-IgA—viral-capsid antigen–IgA; EBV—Epstein–Barr virus. ^a^ Multivariate analysis was performed by unconditional logistic regression models with a constant as the model intercept.

## Data Availability

The data supporting the findings of this study are available on request from the corresponding author.

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
