# Peer review of "Resistin Promotes Nasopharyngeal Carcinoma Metastasis through TLR4-Mediated Activation of p38 MAPK/NF-κB Signaling Pathway"

_cancers, 2022, doi:10.3390/cancers14236003_

Round 1

Reviewer 1 Report

cancers-2026000

The authors analyse the role of resistin in nasopharyngeal cancer based on serum resistin levels and functional assays in vitro and in vivo. The work is of high interest, and the cell biological anaylsis is convincing.

However, there are questions that need to be addressed concerning the analysis of resistin blood levels. Those are given in pairs of numbers (M, Q). This needs to be explained.

Also in tables 1 and 3 it looks like both numbers were statistically analysed in the same procedure together. This analysis has to be done and documented separately.

In tables 2 and 4: explain the line constant.

Author Response

We would like to expression our appreciation to both reviewers for their constructive comments. We have addressed each of the concerns.

Reviewer #1:

Q1) There are questions that need to be addressed concerning the analysis of resistin blood levels. Those are given in pairs of numbers (M, Q). This needs to be explained.

R1: Sorry about this. In this revision, we have made the corrections, adding the explanations in table footnotes.

Q2) Also in tables 1 and 3 it looks like both numbers were statistically analysed in the same procedure together. This analysis has to be done and documented separately.

R2: Indeed, Table 1 presents the general characteristics of NPC cases and the controls, and the differences in resistin levels between NPC patients and the controls. Table 3 presents the differences in resistin levels between NPC patients with different clinical characteristics.

Q3) In tables 2 and 4: explain the line constant.

R3: Our apology, we should explain these annotations in Table 2 and Table 4 are mistake. Indeed, we employed logistic regression analysis with constant as the intercept of the regression model in Table 2 and Table 4. We have corrected this mistake in the revision.

Reviewer 2 Report

Thank you for the opportunity to review this manuscript.

The authors clearly showed that Resistin enhanced nasopharyngeal carcinoma metastasis via TLR4/MAPK/NF-κB signaling.

I think this manuscript well written. And I believe most cancer researchers will be interested in the results.

Author Response

Reviewer #2:

The authors clearly showed that Resistin enhanced nasopharyngeal carcinoma metastasis via TLR4/MAPK/NF-κB signaling. I think this manuscript well written. And I believe most cancer researchers will be interested in the results.

Response: Thanks for the encouraging comment.

Round 2

Reviewer 1 Report

the authors have adequately addressed my concerns